# Semi-Covariance Coefficient Analysis of Spike Proteins from SARS-CoV-2 and Its Variants Omicron, BA.5, EG.5, and JN.1 for Viral Infectivity, Virulence and Immune Escape

**DOI:** 10.3390/v16081192

**Published:** 2024-07-25

**Authors:** Botao Zhu, Huancheng Lin, Jun Steed Huang, Wandong Zhang

**Affiliations:** 1Department of Electrical and Computer engineering, Western University, London, ON N6A 5B9, Canada; bzhu88@uwo.ca; 2School of Information Technology, Carleton University, Ottawa, ON K1S 5B6, Canada; linhuancheng@cunet.carleton.ca; 3Human Health Therapeutics Research Centre, National Research Council of Canada, 1200 Montreal Road, Building M54, Ottawa, ON K1A 0R6, Canada; 4Faculty of Medicine, University of Ottawa, Ottawa, ON K1H 8M5, Canada

**Keywords:** SARS-CoV-2, coronaviruses, variants, spike protein sequence, semi-covariance coefficient, infectivity, virulence, immune escape

## Abstract

Semi-covariance has attracted significant attention in recent years and is increasingly employed to elucidate statistical phenomena exhibiting fluctuations, such as the similarity or difference in charge patterns of spike proteins among coronaviruses. In this study, by examining values above and below the average/mean based on the positive and negative charge patterns of amino acid residues in the spike proteins of SARS-CoV-2 and its current circulating variants, the proposed methods offer profound insights into the nonlinear evolving trends in those viral spike proteins. Our study indicates that the charge span value can predict the infectivity of the virus and the charge density can estimate the virulence of the virus, and both predicated infectivity and virulence appear to be associated with the capability of viral immune escape. This semi-covariance coefficient analysis may be used not only to predict the infectivity, virulence and capability of immune escape for coronaviruses but also to analyze the functionality of other viral proteins. This study improves our understanding of the trend of viral evolution in terms of viral infectivity, virulence or the capability of immune escape, which remains further validated by more future studies and statistical data.

## 1. Introduction

The Fractal DNA hypothesis (FDH) was introduced in 1994 [1]. The arithmetic data derived from DNA sequences, obtained by counting the number of intervening bases from a specific base to the next one (inter-event data), exhibits a dynamical process characterized by the observation of long-range (fractal) correlations. The reports of fractal long-range correlations in DNA sequences were also made by Peng et al. and others [2,3,4]. Unlike the traditional DNA hypothesis, the analysis of RNA and proteins is based on the fragment length between domains with electrical charges. Notably, it underscores the impact on charge behaviors stemming from differences in information reception, expression lengths, or neighbor status, indicating the presence of a fractal structure in stable DNAs [5]. Recent work on FDH in RNA studies [6] involves converting genetic sequences into binary numbers, with purines converted to −1 and pyrimidines converted to +1 [7]. We have previously used the semi-covariance coefficient method to analyze the correlation between the amino acid composition of spike proteins from SARS-CoV-2 and other coronaviruses for viral evolution trends and characteristics associated with fatality or virulence [8]. The semi-covariance is better to elucidate statistical phenomena exhibiting fluctuations, such as the similarity or difference in charge patterns of spike proteins among coronaviruses. The fluctuation analysis method has been previously used in analyzing the long-range correlation of DNA sequences [9]. To further characterize the charge patterns of spike proteins among coronaviruses, we have then used normalized semi-covariance co-efficiency to analyze the charge of the spike protein composition for viral infectivity and virulence [10]. In this study, our objectives are to investigate the analogous electrical charge-specific relationship (+1 for positive amino acids, −1 for negative counterparts, and 0 for neutral ones; the charges of amino acids are determined as listed in the amino acid table in the biochemistry textbook) in the spike protein sequences of coronaviruses for the viral infectivity and virulence [11] and to attempt to associate the viral infectivity and virulence with the viral capability of immune escape. The SARS-CoV-2 virus stands out as one of the longest positive single-stranded RNA viruses [12], and its protein folding, tertiary structure, and functions are intricately linked to the exhibiting of more positive charges than other proteins for binding to or interacting with the receptor [13,14]. Therefore, it is crucial to scrutinize the charge structure/patterns or nonlinear correlation patterns of the spike proteins of SARS-CoV-2 and its variants in comparison to spike proteins from the original SARS-CoV-2 strain, where the negative charge is predominant [15]. This study on viral protein charges furthers our understanding of viral evolution trends in terms of infectivity and virulence and associates viral infectivity and virulence with the ability of viral immune escape.

## 2. Literature Review

### 2.1. Research on Spike Proteins of SARS-CoV-2

The spike protein is a structural protein unique to the surface of coronaviruses. It contains crucial information about the natural evolution of coronaviruses and plays a key role in the viral recognition and invasion of human cells [16,17,18,19,20]. Over the past decade, the spike protein has been one of the most important research subjects in studies of coronaviruses closely related to humans. Following the outbreak of the COVID-19 pandemic, the spike protein quickly became a focal point of research [21]. The authors in [22] utilized pseudo-viruses to monitor the impact of FCS (furin cleavage site)-spike mutations of either Alpha/Omicron, Beta, or Delta variants on viral infectivity and neutralization sensitivity against sera that were drawn from fully vaccinated individuals. In Cordsmeier and colleagues’ study [23], the authors constructed SARS-CoV-2 with a uniform B.1. backbone but with alternative spike proteins to analyze the specific impact of variant spike proteins on infection dynamics. In Emmelot et al.’s study [24], the authors analyzed the functional impact of Omicron BA.4/BA.5 spike mutations on T cell responsiveness to non-conserved epitopes in vaccines. The result showed that several BA.4/BA.5 mutations in the spike protein led to a reduced responsiveness of epitope-specific T cells in subjects that received two doses of an mRNA vaccine based on the ancestral wild-type spike sequence. In the Bains and colleagues’ study [25], the authors assessed the impact of SARS-CoV-2 spike S1-domain glycans and spike proteins from different strains on the ability of pseudotyped lentivirions to undergo DC-SIGN-mediated trans-infection. In Escalera et al.’s study [26], the authors analyzed a set of emerging SARS-CoV-2 variants to investigate how different sets of mutations may impact spike protein processing, and they demonstrated that the mutations in spike protein present in these variants that become epidemiologically prevalent in humans are linked to an increase in spike protein processing and virus transmission. The authors [27] introduced a total of 48 mutations in the spike protein of SARS-CoV-2 variants and demonstrated that several amino acid changes found in Omicron spike proteins impair infectivity. Additionally, numerous alterations in the N-terminal domain (NTD) and receptor-binding domain (RBD) of BA.1 and/or BA.2 spike proteins impact neutralization by sera from individuals vaccinated with BNT/BNT-based vaccines and therapeutic antibodies.

### 2.2. Research on Electrostatic Feature of SARS-CoV-2 Spike Protein

The spike protein of SARS-CoV-2 is made up of amino acids, each with specific chemical properties. Amino acids like arginine (Arg) and lysine (Lys) have positively charged side chains at physiological pH (around neutral pH). These positively charged residues can be found within different regions of the spike protein, including the RBD and other functional domains. These charges play critical roles in viral attachment, entry, and immune recognition, making them important factors in understanding and combating COVID-19 [28,29,30,31]. The increase in positive charge on spike protein of SARS-CoV-2 variants (such as Omicron) altered the biochemical properties of spike protein and may influence virion survival and promote transmission [32]. Studies found that mutations increased the electrostatic interactions of the Omicron spike protein RBD with ACE2 and promoted infectivity and transmission of the variants [33,34]. Nguyen and colleagues [35] investigated the evolving positive charge of the SARS-CoV-2 spike protein and found that the Omicron variant has enhanced binding rates to negatively charged glycocalyx and significantly stronger interactions with heparan sulfate as compared to Delta. This increased dependence on the heparan sulfate for viral attachment and infection suggests new therapeutic and diagnostic opportunities. Kim et al. [36] compared the spike proteins of SARS-CoV-2 Omicron sublineages with earlier SARS-CoV-2 variants and found that the SARS-CoV-2 Omicron has a higher electric field line density than that of earlier SARS-CoV-2 variants, which indicates a stronger interaction between the Omicron spike protein and the ACE2 receptor. Božič and Podgornik [37] found that the trend of increase in the positive charge on spike protein of SARS-CoV-2 variants has been halted with the appearance of Omicron variants, while these sublineages display a greater diversity in their composition of ionizable amino acids. Pascarella and colleagues [38] examined the spike proteins from B.1.617.1 (Kappa), B.1.617.2 (Delta), and B.1.617.3 and found that these variants exhibit significant changes in the spike protein’s surface electrostatic potential, particularly in Delta, which may enhance the interaction with the negatively charged ACE2 receptor and increase viral transmission. Lu and colleagues [39] calculated the charge distributions of SARS-CoV, SARS-CoV-2, and variants of concern using net charge calculation formulas and found that the SARS-CoV-2 spike protein had more positive charges than that of SARS-CoV. Further analysis showed that variants, particularly the Delta variant, had even higher positive charges in the S1 domain, significantly increasing Coulomb’s force with the negatively charged ACE2 receptor and potentially leading to higher infectivity. Another study [40] investigated the main electrostatic features involved in the interaction between the RBD of the SARS-CoV-2 spike protein and the human receptor ACE2. Using the FORTE approach, which models proton fluctuations and computes free energies for many protein-protein systems, these authors analyzed wild-type and critical variants, focusing on pH-dependent binding affinities, protein charges, charge regulation capacities, and dipole moments. They revealed a linear correlation, termed the “RBD charge rule”, between the total charge of the RBD and its binding affinity to ACE2, providing a quick test for predicting the severity of future SARS-CoV-2 variants.

## 3. Methodologies

The coronaviral spike protein and RNA sequences used in this research were obtained from NCBI GenBank and GISAID databases, including Wuhan strain SARS-CoV-2, UK variant (B.1.1.7), Delta variant (B.1.617), Omicron, and Omicron subvariants BA.5 (B.1.1.529), EG.5, and JN.1. The viral RNA sequences obtained from the GISAID database were translated into protein sequences before analysis. As said above, a positive-charged amino acid is represented as +1, a negative-charged amino acid is represented as −1, and a neutral amino acid is represented as 0. Assuming two distinct viral protein sequences are converted into sequences or symbols represented as charges, labeled as $X=(x_{1},\ldots ,x_{n})$ and $Y=(y_{1},\ldots ,y_{n})$, with *X* serving as the baseline, various computational methods are proposed to compare and analyze the correlation between viral protein sequences from multiple dimensions, aiming to better understand the evolution trend of the virus, such as infectivity and virulence. These computational methods are detailed below.

### 3.1. Pearson Correlation Coefficient

The Pearson correlation coefficient quantifies the linear correlation between two variables, which is calculated as the ratio of the covariance of two variables to the product of their standard deviations. It can also be used to represent the correlation between two viral sequences, which is given by [41]
(1)$$\rho \left(X,Y\right)=\frac{E\left[\left(X-E[X]\right)\left(Y-E[Y]\right)\right]}{\sqrt{E\left[X^{2}\right]-{\left(E\left[X\right]\right)}^{2}}\sqrt{E\left[Y^{2}\right]-{\left(E\left[Y\right]\right)}^{2}}}.\;$$

### 3.2. Semi-Variance Correlation

#### 3.2.1. Semi-Variance Correlation Coefficient

While the Pearson correlation coefficient excels at expressing the correlation among multiple variables, it is limited to capturing only linear correlations and overlooks many other types of relationships or correlations. Inspired by the Pearson correlation coefficient, we propose the semi-covariance coefficient, which can measure the nonlinear correlation between variables and provide more detailed information and insights. As shown in Figure 1, when employing Pearson correlation to measure the relationship between two sequences, it only indicates the linear relationship (positive and negative correlation folded together) between the sequences. However, when using semi-variance correlation to measure the relationship between two sequences, it reveals the nonlinear relationship (positive and negative correlation unfolded separately) between the sequences, which can better uncover the relationship between them.

To shift from linear correlation to nonlinear correlation, the Rectified Linear Unit (ReLU) [42] [https://en.wikipedia.org/wiki/Rectifier_(neural_networks) (accessed on 21 May 2024)] is employed in this study. ReLU is an activation function that introduces nonlinearity, allowing the model to capture or unfold and represent more complex relationships in the data. The basic form of ReLU is given by the following:(2)$$ReLU\left(R\right)=\max\left(0,R\right),$$
where *R* is a real number. Then, *R* can be expressed or unfolded as following:(3)$$R=ReLU\left(R\right)-ReLU\left(-R\right),$$
and the Pooling of the expectation of *R*
(4)$$E[R]=E[ReLU\left(R\right)-ReLU\left(-R\right)].$$

Setting R=(X−E[X])(Y−E[Y]) and substituting it into Equation (1), the semi-covariance coefficient is thus defined as following:(5)$$S_{pos}\left(X,Y\right)=\frac{E[ReLU((X-E[X])(Y-E[Y]))]}{\sqrt{E\left[X^{2}\right]-{\left(E\left[X\right]\right)}^{2}}\sqrt{E\left[Y^{2}\right]-{\left(E\left[Y\right]\right)}^{2}}},$$
(6)$$S_{neg}\left(X,Y\right)=\frac{E\left[ReLU\left(-\left(X-E[X]\right)\left(Y-E[Y]\right)\right)\right]}{\sqrt{E\left[X^{2}\right]-{\left(E\left[X\right]\right)}^{2}}\sqrt{E\left[Y^{2}\right]-{\left(E\left[Y\right]\right)}^{2}}},$$
where $S_{pos}$ is the positive correlation covariance coefficient or the convergent part, and $S_{neg}$ is the negative correlation covariance coefficient or the divergent part. According to the mean of each two variables, all variables are divided or unfolded into four quadrants. $S_{pos}$ belongs to the first and third quadrants, while $S_{neg}$ belongs to the second and fourth quadrants. We also include the following indicators to Semi-variance correlation to assess its advantage.

#### 3.2.2. Quantity of Charge

The quantity of charge of the sequence *Y* is
(7)$$Q_{Y}=\sum_{i=1}^{n}y_{i}.$$

#### 3.2.3. Gravity

As ReLU(R) is a sequence, denoted as $\left(ReLU\left(r_{1}\right),\cdots ReLU(r_{n})\right)$, where each is assumed to be the weight, the gravity of converge part of *Y* is calculated by
(8)$$\begin{array}{cc} C_{pos} & =Round\left(\frac{\sum_{i}^{n}ReLU\left(r_{i}\right)I_{i}}{\sum_{i}^{n}ReLU\left(r_{i}\right)}\right)\\ \; & =Round\left(\frac{\sum_{i}^{n}ReLU({(x}_{i}-E[Y])(y_{i}-E[Y]))I_{i}}{\sum_{i}^{n}ReLU({(x}_{i}-E[Y])(y_{i}-E[Y]))}\right), \end{array}$$
where $I_{i}$ is the position of $ReLU(r_{i})$ in the sequence, and *Round* is the rounding operation. Likewise, the gravity of divergent part of *Y* is given by
(9)$$\begin{array}{cc} C_{neg} & =Round\left(\frac{\sum_{i}^{n}ReLU\left(-r_{i}\right)I_{i}}{\sum_{i}^{n}ReLU\left({-r}_{i}\right)}\right)\\ \; & =Round\left(\frac{\sum_{i}^{n}ReLU({-(x}_{i}-E[Y])(y_{i}-E[Y]))I_{i}}{\sum_{i}^{n}ReLU({-(x}_{i}-E[Y])(y_{i}-E[Y]))}\right). \end{array}$$

#### 3.2.4. Charge Span

The charge span of *Y* is given by
(10)$$M_{Y}=\left|C_{pos}-C_{neg}\right|.$$

#### 3.2.5. Reproduction Rate

The reproduction rate of *Y* is calculated by
(11)$$O_{Y}=\frac{M_{Y}O_{X}}{M_{X}},$$
where $O_{X}$ and $M_{X}$ are the reproduction rate and the charge span of virus *X*, respectively.

#### 3.2.6. Maximal Position

The position of maximal value in $S_{pos}$ is
(12)$$P_{pos}\;\in {argmax}_{1\leq i\leq n}S_{pos}\left(i\right).$$

Likewise, the position of maximal value in $S_{neg}$ is
(13)$$P_{neg}\;\in {argmax}_{1\leq i\leq n}S_{neg}\left(i\right).\;$$

#### 3.2.7. Charge Density

The charge density of *Y* is calculated by
(14)$$D_{Y}=\frac{P_{pos}}{P_{neg}O_{Y}}.$$

#### 3.2.8. Virulence

The virulence of *Y* is calculated by
(15)$$V_{Y}=\frac{D_{Y}V_{X}}{D_{X}},$$
where $V_{X}$ and $D_{X}$ are the virulence and the charge density of *X*, respectively.

## 4. Result and Discussion

To assess the correlation between spike protein sequences of SARS-CoV-2 and its variants using the proposed methods, the Wuhan strain SARS-CoV-2 is employed as the sequence baseline, and the Delta variant is used as the data point reference. We first examine the difference between Pearson correlation and semi-variance correlation in measuring the correlation between BA.5 and Wuhan SARS-CoV-2. As shown in Figure 2a, Pearson correlation can only represent the relationship between these two viral spike protein sequences within the first quadrant of the coordinate axis. However, in Figure 2b, Semi-variance correlation can demonstrate the correlation between these two viral sequences in all four quadrants (unfolded with ReLU function). This result indicates that the proposed semi-variance correlation can uncover the multiple nonlinear relationships between viral sequences, thus facilitating the inference of the evolution between sequences.

The comparison results of different variants are presented in Table 1. The charge span $M_{Y}$ reflects the infectivity of viruses. The higher the $M_{Y}$ value of the viral spike protein, the greater the viral infectivity. The results are consistent with the actual infectivity of the virus and its variants. For instance, Omicron and EG.5 were both major prevalent variants a couple of years ago, while JN.1 is currently the predominant variant and more infectious. EG.5 is more infectious ($M_{Y}$ value of 551) than Omicron ($M_{Y}$ value of 270), while JN.1 is more infectious ($M_{Y}$ value of 565) than EG.5 (Table 1). Furthermore, the charge density $D_{Y}$ represents viral virulence. The higher the value of $D_{Y}$, the greater the virulence of the virus. Due to the wide campaign of vaccination against SARS-CoV-2 and its variants and the establishment of herd immunity to the virus and its variants, the viral virulence is minimized in the immunized population. The value of $D_{Y}$ may thus correlate with the capability of immune escape for the variants due to the rapid evolution of the variants. Immune escape is one aspect of the evolution of viral variants, and new variants gain more and more capabilities of immune escape [43]. People who received vaccinations or recovered from infections gain immunity or cross-immunity against new variants of the virus. For the survival or evolution of the virus, its new variants or new mutants have to develop the ability of immune escape to evade T cells, innate immunity and population immunity [43]. Mutations in the viral spike proteins allow new variants to emerge with the capabilities of greater transmissibility and immune escape [44]. The values of both predicted infectivity and virulence may thus correlate with the capability of immune escape for these variants. Although the new variants develop a stronger immune escape capability, the cross-immune response established previously in the human body may relieve the symptoms of people who are infected with the new variants. Our analysis results appear to be consistent with the trends of viral infectivity and virulence associated with enhanced immune escape.

## 5. Conclusions

In this study, we examined the positive charge patterns of spike proteins from SARS-CoV-2 and its current circulating variants. Our analyses indicate that the charge span (*M_Y_*) value may predict the infectivity of the viral variants, with a high charge span value exhibiting high infectivity. The charge density (*D_Y_*) may predict the virulence of the viral variants, with a high charge density value displaying more virulence for the original strain of SARS-CoV-2 or more capability of immune escape for current circulating variants derived from Omicron. Our analyses provide a profound understanding of the nonlinear pattern of viral evolution trend and identify the protein mutation characteristics that may be associated with viral infectivity and virulence for developing the immune escape capability of current circulating variants and for predicting the evolving trend of future new variants. More future studies and statistical data will be needed to further validate our findings and the association. This semi-covariance coefficient analysis may be used not only to predict the infectivity, virulence and capability of immune escape for coronaviruses but also to analyze the functionality of other viral proteins. There are limitations to our study. The charges of the amino acids may change under different pH environments, such as more acidic or more basic conditions. These conditions were not considered in our study since the pH is generally maintained from pH 7.35 to pH 7.45 in the human body. The infectivity and virulence of the viruses on the human body are affected by many factors, such as underlying diseases (hypertension, heart and lung diseases, medications of the patients, etc.), vaccinations, immunity of the affected individuals, etc. These conditions were not considered in the study. Another limitation is the Equations of (10) to (15) in our study. To further validate Equations (10) to (15), more research and statistical data are needed in the future.

## Figures and Tables

**Figure 1 viruses-16-01192-f001:**
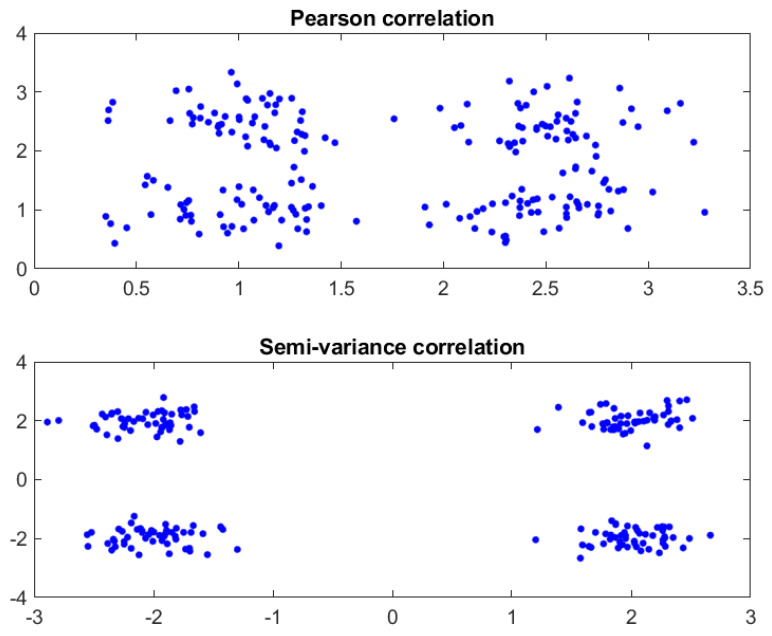
Visual comparison of Pearson correlation and Semi-variance correlation.

**Figure 2 viruses-16-01192-f002:**
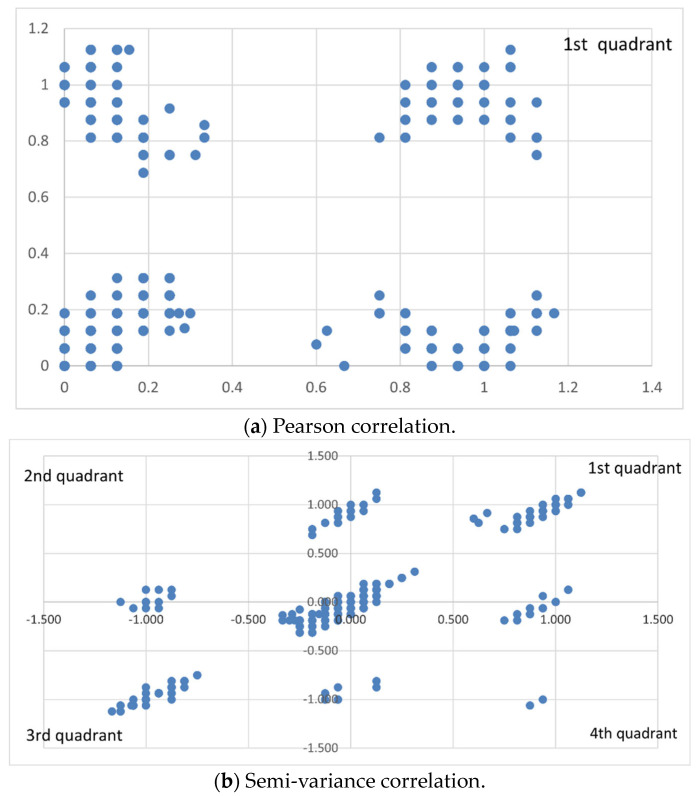
Visual comparison of correlations between the spike protein sequences of Wuhan SARS-CoV-2 and BA.5 variant using (**a**) Pearson correlation and (**b**) semi-variance correlation. The analyses with semi-variance correlation show multiple nonlinear patterns of viral evolution.

**Table 1 viruses-16-01192-t001:** Comparison of spike proteins of SARS-CoV-2 variants *.

	UK (B.1.1.7)	Delta (B.1.617)	Omicron	BA.5	EG.5	JN.1
$S_{pos}$	99.58%	99.44%	98.09%	96.04%	87.63%	63.9%
$S_{neg}$	0.32%	0.17%	1%	0.4%	1.95%	7.75%
ρ	0.9926	0.9927	0.9709	0.9564	0.8559	0.5615
$Q_{Y}$	17	15	18	18	17	19
$C_{pos}$	626	629	631	665	718	868
$C_{neg}$	655	557	361	568	167	303
***** $M_{Y}$	**29**	**72**	**270**	**97**	**551**	**565**
$O_{Y}$	0.84	2.08	7.79	2.8	15.89	16.3
$P_{pos}$	516	518	518	1264	1262	1262
$P_{neg}$	614	616	216	683	97	97
***** $D_{Y}$	**1**	**0.4**	**0.31**	**0.66**	**0.82**	**0.8**
$V_{Y}$	5.33%	2.15%	1.64%	3.51%	4.35%	4.24%
Actual rate of death (sourced from the Internet)	1.3–5.3%	0.3–3.4%	0.06–0.3%	0.06–0.3%	/	1.81%

* The charge span $M_{Y}$ reflects the infectivity of viruses and the variants (higher value represents high infectivity). The charge density $D_{Y}$ represents the virulence of viruses and the variants. Both viral infectivity and virulence further evolve to correlate with viral immune escape.

## Data Availability

The calculation formulas are provided in this paper, and data will be available upon request.

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
