# Peer review of "Semi-Covariance Coefficient Analysis of Spike Proteins from SARS-CoV-2 and Its Variants Omicron, BA.5, EG.5, and JN.1 for Viral Infectivity, Virulence and Immune Escape"

_viruses, 2024, doi:10.3390/v16081192_

Round 1

Reviewer 1 Report

Comments and Suggestions for Authors

In this communication, the authors used semi-variance correlation analysis to investigate the charge patterns of spike proteins from SARS-CoV-2 strain and its circulating variants. The contents of this work are potentially interesting. However, the manuscript should be major revised to address the following issues:

1)      It is not clear how the charges of amino acids in the spike protein are determined! Which approaches are used to determine them, classically or quantum mechanically? How did the authors handle the amino acids that are influenced by different pH environments?

2)      The communication lacks relevant references, particularly those that adopt a different approach, such as:

https://doi.org/10.1093/bioadv/vbae053

https://doi.org/10.1016/j.isci.2023.106230

https://doi.org/10.1021/acs.jpclett.2c00423

https://doi.org/10.3390/ijms23052870

https://doi.org/10.1093/ve/vead040

DOI: 10.2147/IDR.S342068

3)     A few equations lack a solid basis, particularly equations 10 to 15. Authors must support the validity of these equations based on additional research or groups to make the conclusion more solid. Particularly, the conclusions drawn from this communication depend primarily on these equations, and without appropriate evidence of the validity of these equations, this manuscript will lose its importance.

Similarly, a sentence such as “This enhances our understanding in deciphering viral evolution in terms of its infectivity, virulence, or the capability of immune escape” requires more evidence.

4)      Limitations of the work should be mentioned in the result and discussion section.

Comments on the Quality of English Language

The manuscript is well written, but minor editing of the English language required

Author Response

In this communication, the authors used semi-variance correlation analysis to investigate the charge patterns of spike proteins from SARS-CoV-2 strain and its circulating variants. The contents of this work are potentially interesting. However, the manuscript should be major revised to address the following issues:

1)      It is not clear how the charges of amino acids in the spike protein are determined! Which approaches are used to determine them, classically or quantum mechanically? How did the authors handle the amino acids that are influenced by different pH environments?

Response: Thanks very much for the comments. The charges of the amino acids are accounted based on the amino acid chart in textbooks which is usually determined at neutral pH (pH7). The negative charge of an amino acid is accounted as -1, and the positive charge of an amino acid is accounted as +1. The viruses are produced and acted in human body, and the pH of the human body fluids is mostly buffered between pH 7.35 and 7.45. The charges of the amino acids are essentially the same as they are at neutral pH condition. Since the infectivity and virulence caused by the viruses occur in human body fluids, we did not consider the changes of amino acid charges in different pH environments.

2)      The communication lacks relevant references, particularly those that adopt a different approach, such as:

https://doi.org/10.1093/bioadv/vbae053

https://doi.org/10.1016/j.isci.2023.106230

https://doi.org/10.1021/acs.jpclett.2c00423

https://doi.org/10.3390/ijms23052870

https://doi.org/10.1093/ve/vead040

DOI: 10.2147/IDR.S342068

Response: Thanks very much for the comments and the references. These references and studies have been cited in the revised manuscript.

3)     A few equations lack a solid basis, particularly equations 10 to 15. Authors must support the validity of these equations based on additional research or groups to make the conclusion more solid. Particularly, the conclusions drawn from this communication depend primarily on these equations, and without appropriate evidence of the validity of these equations, this manuscript will lose its importance.

Similarly, a sentence such as “This enhances our understanding in deciphering viral evolution in terms of its infectivity, virulence, or the capability of immune escape” requires more evidence.

Response: Thanks very much for the comments which are important.  The validity of the equations requires actual data to validate. The results obtained from our previous studies (Huang JS et al. 2021; Xu T et al. 2023) for viral infectivity and virulence are essentially the same for the viral infectivity and virulence reported in the literature. It is expected that more studies and data will be reported, which will further validate the equations in our study. The sentence has been revised in the manuscript “our study may improve the understanding of viral evolution in terms of its infectivity, virulence or the capability of immune escape; however, more future studies and statistical data are required to further validate our study”.

4)      Limitations of the work should be mentioned in the result and discussion section.

Response: There are limitations of the study. The charges of the amino acids may change under different pH environments, such as more acidic or more basic conditions. These conditions were not considered in this study. The infectivity and virulence of the viruses on human body are affected by many factors, such as underlying diseases (hypertension, heart and lung diseases, medications of the patients), vaccinations and immunity of the affected individuals, etc. These conditions were not considered in the study. The equations of 10-15 require more future studies and statistical data to validate further.

The limitations of our study have been incorporated into the Discussion of the revised manuscript.

Reviewer 2 Report

Comments and Suggestions for Authors

Journal: Viruses

Manuscript number: 3045225

Title: Semi-covariance coefficient analysis of spike proteins from 2 SARS-CoV2 and its variants Omicron, BA.5, EG.5, and JN.1 for 3 viral infectivity, virulence and immune escape

 In this study, by examining values above and below the average/mean based on the positive and negative charge patterns of amino acid residues in the spike proteins of SARS-CoV-2 and its current circulating variants, the proposed methods offer profound insights into the nonlinear evolving trends of those viral spike proteins.

The study indicates that the charge span value can predict the infectivity of the virus, and the charge density can estimate the virulence of the virus, and both predicated infectivity and virulence appears to be associated with the capability of viral immune escape. This semi-covariance coefficient analysis may be used not only to predict the infectivity, virulence and the capability of immune escape for coronaviruses but also to analyze the functionality of other viral proteins.

The objectives were to investigate the analogous electrical charge-specific relationship (+1 for positive amino acids, -1 for negative counterparts, 0 for neutral ones) in the spike protein sequences of coronaviruses for the viral infectivity and virulence [7] and to attempt associating the viral infectivity and virulence with the viral capability of immune escape.

Authors have used normalized semi-covariance co-efficiency to analyze the charge of the spike protein composition for the viral infectivity and virulence [6].

MAJOR

1. Abstract: Complex modeling has attracted significant attention in recent years and is increasingly 23 employed to elucidate statistical phenomena exhibiting fluctuations, such as the similarity or difference in charge patterns of spike proteins among coronaviruses.

No modeling is done in this study.

2. In Introduction: “The Fractal DNA hypothesis (FDH) was introduced in 1992 [1].

[1] Peters, E.E. Fractal DNA Analysis: Applying Chaos Theory to Investment and Economics; Wiley: Hoboken, NJ, USA, 1994.

3. The first reports of fractal long-range correlations in DNA sequences were obtained by:

C.-K. Peng, S.V. Buldyrev, A.L. Goldberger, S. Havlin, F. Sciortino, M. Simons, H.E. Stanley, Long-range correlations in nucleotide sequences, Nature 356 (1992) 168–170.

S.V. Buldyrev, A.L. Goldberger, S. Havlin, R.N. Mantegna, E. Matsa, C.-K. Peng, M. Simons, H.E. Stanley, Long-range correlations properties of coding and noncoding DNA sequences: GenBank analysis, Phys. Rev. E 51 (1995) 5084–5091.

R. Voss, Evolution of long-range fractal correlations and 1/f noise in DNA base sequences, Phys. Rev. Lett. 68 (1992) 3805–3808.

4. Given the non-linearity of DNA sequences and the non-stationarity the Detrending fluctuation analysis was developed:

C.-K. Peng, S.V. Buldyrev, S. Havlin, M. Simons, H.E. Stanley, A. Goldberger, Mosaic organization of DNA nucleotides, Phys. Rev. E 49 (1994) 1685–1689.

5. Line 64: A reference is needed in:

“…comparison to spike proteins from original SARS-CoV-2 strain, where the negative charge is predominant”

6. Line 82: “It is also can…, replaced by “It can also be…”

7. Importantly, I did not understand Lines 101 to 149.

Consequently, I could not follow the Results and Discussion.

The results must be taken at face value.

8. The term Z can easily be confounded with the symbol commonly used to represent integers.

The sequences X and Y are formed by 1, -1, and 0. Z is a real number.

9. In Discussion it is stated that:

The charge density may predict the virulence of the viral variants, with having a high charge density value displaying more virulence for the original strains of SARS-CoV-2 or more capability of immune escape for current circulating variants derived from Omicron.”

The reader is left with the question whether the charge density is the cause or is associated to more virulence; and/or it permits that SARS-CoV-2 can escape from the immune system.

Recall that: “Semi-variance correlation coefficient measures the non-linear correlation between variables.”

It is also ambiguous to claim that:

“Our analyses provide a profound understanding of the nonlinear pattern of viral evolution…”

What and how is the nonlinear pattern of viral evolution?

Comments on the Quality of English Language

Author Response

Manuscript number: 3045225

Title: Semi-covariance coefficient analysis of spike proteins from 2 SARS-CoV2 and its variants Omicron, BA.5, EG.5, and JN.1 for 3 viral infectivity, virulence and immune escape

In this study, by examining values above and below the average/mean based on the positive and negative charge patterns of amino acid residues in the spike proteins of SARS-CoV-2 and its current circulating variants, the proposed methods offer profound insights into the nonlinear evolving trends of those viral spike proteins.

The study indicates that the charge span value can predict the infectivity of the virus, and the charge density can estimate the virulence of the virus, and both predicated infectivity and virulence appears to be associated with the capability of viral immune escape. This semi-covariance coefficient analysis may be used not only to predict the infectivity, virulence and the capability of immune escape for coronaviruses but also to analyze the functionality of other viral proteins.

The objectives were to investigate the analogous electrical charge-specific relationship (+1 for positive amino acids, -1 for negative counterparts, 0 for neutral ones) in the spike protein sequences of coronaviruses for the viral infectivity and virulence [7] and to attempt associating the viral infectivity and virulence with the viral capability of immune escape.

Authors have used normalized semi-covariance co-efficiency to analyze the charge of the spike protein composition for the viral infectivity and virulence [6].

MAJOR

  1. Abstract: Complex modeling has attracted significant attention in recent years and is increasingly 23 employed to elucidate statistical phenomena exhibiting fluctuations, such as the similarity or difference in charge patterns of spike proteins among coronaviruses.

No modeling is done in this study.

Response: Thanks very much for the comments. The wording has been changed to semi-covariance.

  1. In Introduction: “The Fractal DNA hypothesis (FDH) was introduced in 1992 [1].

[1] Peters, E.E. Fractal DNA Analysis: Applying Chaos Theory to Investment and Economics; Wiley: Hoboken, NJ, USA, 1994.

Response: The year has been corrected to 1994 in the revised manuscript.

  1. The first reports of fractal long-range correlations in DNA sequences were obtained by:

C.-K. Peng, S.V. Buldyrev, A.L. Goldberger, S. Havlin, F. Sciortino, M. Simons, H.E. Stanley, Long-range correlations in nucleotide sequences, Nature 356 (1992) 168–170.

S.V. Buldyrev, A.L. Goldberger, S. Havlin, R.N. Mantegna, E. Matsa, C.-K. Peng, M. Simons, H.E. Stanley, Long-range correlations properties of coding and noncoding DNA sequences: GenBank analysis, Phys. Rev. E 51 (1995) 5084–5091. 

  1. Voss, Evolution of long-range fractal correlations and 1/f noise in DNA base sequences, Phys. Rev. Lett. 68 (1992) 3805–3808.

Response: Thanks very much for the references. These references have been cited in the revised manuscript.

  1. Given the non-linearity of DNA sequences and the non-stationarity the Detrending fluctuation analysis was developed:

C.-K. Peng, S.V. Buldyrev, S. Havlin, M. Simons, H.E. Stanley, A. Goldberger, Mosaic organization of DNA nucleotides, Phys. Rev. E 49 (1994) 1685–1689.

Response: Thanks very much for the comment and the reference. The comment and reference have been addressed and cited in the revised manuscript.

  1. Line 64: A reference is needed in:

“…comparison to spike proteins from original SARS-CoV-2 strain, where the negative charge is predominant”

Response: Thanks for the comment. A reference paper for the original Wuhan strain of SARS-CoV-2 virus has been cited in the revised manuscript.

  1. Line 82: “It is also can…, replaced by “It can also be…”

Response: Thanks for finding the typo. It has been corrected in the revised manuscript.

  1. Importantly, I did not understand Lines 101 to 149.

Consequently, I could not follow the Results and Discussion.

The results must be taken at face value.

Response: Thanks for the comments. This part has been revised. We hope the revisions are satisfactory.

Equations (2)(3)(4) are operations related to ReLU. ReLU is used in this research to unfold and represent complex relationships between virus sequences. Please refer to https://en.wikipedia.org/wiki/Rectifier_(neural_networks) for more details about ReLU.

Equations (5)(6) are our proposed semi-covariance coefficient to analyze the relatedness of viral sequences.

Equations (8)(9) calculate the gravity of the sequences because every sequence, such as a virus sequence, has a gravity.

Equations (7)(10)(11)(12)(13)(14)(15) calculate the correlations of the sequences in various dimensions according to the proposed semi-variance correlation coefficient.

The experimental results are all analyzed using the above equations to analyze the relationships between virus sequences.

  1. The term Z can easily be confounded with the symbol commonly used to represent integers.

The sequences X and Y are formed by 1, -1, and 0. Z is a real number.

Response: Thanks for the comments. We have used R instead of Z in the revised manuscript.

  1. In Discussion it is stated that:

The charge density may predict the virulence of the viral variants, with having a high charge density value displaying more virulence for the original strains of SARS-CoV-2 or more capability of immune escape for current circulating variants derived from Omicron.”

The reader is left with the question whether the charge density is the cause or is associated to more virulence; and/or it permits that SARS-CoV-2 can escape from the immune system.

Recall that: “Semi-variance correlation coefficient measures the non-linear correlation between variables.”

It is also ambiguous to claim that:

“Our analyses provide a profound understanding of the nonlinear pattern of viral evolution…”

What and how is the nonlinear pattern of viral evolution?

Response: Thanks very much for the comments. Our results suggest that the charge density of the viral spike protein is associated with more virulence or/and associated with the viral capability of immune escape. For nonlinear pattern of viral evolution, Figure 2b shows such a nonlinear pattern of the viral evolution.    

Round 2

Reviewer 1 Report

Comments and Suggestions for Authors

The authors have adequately addressed most of my comments. The revised version has been significantly improved. Therefore, I have no further comments.

Comments on the Quality of English Language

There are some typos, so a minor English edit is required.

Author Response

Thanks very much. The typos have been corrected and the English has been further edited for better flow. 

Reviewer 2 Report

Comments and Suggestions for Authors

My concerns were answered.

I have only one comment: 

The following sentence is unfortunate (lines 312-313):

“The equations of 10 to 15 of our study require more future studies and statistical data to validate further.”

I suggest either to delete it or to rewrite it as:

“To further validate equations 10 to 15 of our study, more research and statistical data are needed in the future.”

If the equations require validation, then the results also require validation.

Authors converted spike sequences of amino acids into sequences of -1, 0, 1 according to the electric charge of the amino acid.  I did not see a self-similarity of the sequences or a measure of the fractal dimension. The semi-covariance does not unfold the mechanisms of evolution.

Comments on the Quality of English Language

none

Author Response

Responses to Reviewer #2 comments:

My concerns were answered.

I have only one comment: 

The following sentence is unfortunate (lines 312-313):

“The equations of 10 to 15 of our study require more future studies and statistical data to validate further.”

I suggest either to delete it or to rewrite it as:

“To further validate equations 10 to 15 of our study, more research and statistical data are needed in the future.”

If the equations require validation, then the results also require validation.

Response: The sentence has been revised in the manuscript according to the suggested.

Authors converted spike sequences of amino acids into sequences of -1, 0, 1 according to the electric charge of the amino acid.  I did not see a self-similarity of the sequences or a measure of the fractal dimension. The semi-covariance does not unfold the mechanisms of evolution.

Response: Thanks for the comment. The viral spike protein is mutated to have more positively-charged amino acids which allows the virus to be more infectious and transmissible. This is one aspect of viral evolution which will be reflected to be more +1 (positively-charged amino acids) accounted in our calculation. In this case, our analysis with the semi-covariance shows the trend of viral evolution. We did not claim that the semi-covariance analysis unfolds the mechanism of viral evolution. We have changed the wording in the revised manuscript that the study improves our understanding on the trend of viral evolution.
